# Priming of Resistance-Related Phenolics: A Study of Plant-Associated Bacteria and *Hymenoscyphus fraxineus*

**DOI:** 10.3390/microorganisms9122504

**Published:** 2021-12-02

**Authors:** Greta Striganavičiūtė, Jonas Žiauka, Vaida Sirgedaitė-Šėžienė, Dorotėja Vaitiekūnaitė

**Affiliations:** Laboratory of Forest Plant Biotechnology, Institute of Forestry, Lithuanian Research Centre for Agriculture and Forestry, Liepų Str. 1, Girionys, LT-53101 Kaunas, Lithuania; greta.striganaviciute@lammc.lt (G.S.); jonas.ziauka@lammc.lt (J.Ž.); doroteja.vaitiekunaite@lammc.lt (D.V.)

**Keywords:** *Pseudomonas*, *Paenibacillus*, *Fraxinus* *excelsior*, *Hymenoscyphus fraxineus*, phenols, flavonoids, carotenoids, chlorophyll, pathogenic resistance

## Abstract

European ash (*Fraxinus excelsior*) is highly affected by the pathogenic fungus *Hymenoscyphus fraxineus* in all of Europe. Increases in plant’s secondary metabolite (SM) production is often linked tol enhanced resistance to stress, both biotic and abiotic. Moreover, plant-associated bacteria have been shown to enhance SM production in inoculated plants. Thus, our hypothesis is that bacteria may boost ash SM production, hence priming the tree’s metabolism and facilitating higher levels of resilience to *H. fraxineus*. We tested three different ash genotypes and used *Paenibacillus* sp. and *Pseudomonas* sp. for inoculation in vitro. Total phenol (TPC), total flavonoid (TFC) and carotenoid contents were measured, as well as the chlorophyll *a/b* ratio and morphometric growth parameters, in a two-stage trial, whereby seedlings were inoculated with the bacteria during the first stage and with *H. fraxineus* during the second stage. While the tested bacteria did not positively affect the morphometric growth parameters of ash seedlings, they had a statistically significant effect on TPC, TFC, the chlorophyll *a/b* ratio and carotenoid content in both stages, thus confirming our hypothesis. Specifically, in ash genotype 64, both bacteria elicited an increase in carotenoid content, TPC and TFC during both stages. Additionally, *Pseudomonas* sp. inoculated seedlings demonstrated an increase in phenolics after infection with the fungus in both genotypes 64 and 87. Our results indicate that next to genetic selection of the most resilient planting material for ash reforestation, plant-associated bacteria could also be used to boost ash SM production.

## 1. Introduction

Under modern climate change conditions, there is a massive weakening of trees due to the spread of pests and diseases in forests, which result in huge economic, genetic and biodiversity losses. Therefore, more attention should be paid to the increase in forest resistance and productivity as they are associated with the negative effects of pathogens, by selecting tree genotypes that are resistant to these disadvantageous factors [1,2,3,4].

European ash (*Fraxinus excelsior* L.) is a valuable hardwood species in forest ecosystems as well as in the open landscape. Furthermore, this tree species is important due to its economic value and related biodiversity [5,6,7]. Over the last 20 years, an epidemic of ash dieback (ADB), caused by the fungal pathogen *Hymenoscyphus fraxineus* (syn. *Chalara fraxinea*/*H. pseudoalbidus*), has spread throughout Europe and is drastically reducing European ash populations [8]. According to the data of the State Forest Service, European ash is a vulnerable tree species in Lithuanian forests. Data show that, in 2020, 13% of the total forest trees were damaged by diseases and 34.15% of them suffered from *H. fraxineus* [9]. This disease leads to widespread dieback and tree mortality, therefore, most concerningly, it may result in a substantial reduction or even loss of *F. excelsior* in most of Europe [8,10,11]. Thus, due to the rapid and widespread occurrence and intensification of this disease, the management of European ash has become increasingly challenging.

Research indicates that bacteria and fungi, which live in the plant episphere and endosphere, can directly decrease the negative impact of invaders through resource competition, parasitism and antibiosis, or, indirectly, by activating the plant’s defense system [12,13]. Moreover, some endophytic microorganisms are able to promote plant growth and development, resulting in an increased resilience in response to stress and pathogens [14,15]. These processes can be caused by microorganisms’ interaction with their host’s metabolism and environmental conditions [16,17]. Plant-growth-promoting bacteria are used as an eco-friendly alternative to chemical fertilizers and pesticides in agriculture [18,19]. However, studies of culturable bacteria and their antagonistic potential in forestry are still lacking [20,21].

Several studies have reported that 1–5% of European ash genotypes may possess a durable, high, but partial resistance to *H. fraxineus* [11,22,23,24,25,26,27]. Tree resistance to pathogens is related to their ability to synthesize and mobilize secondary metabolites (SMs) [2,28,29,30,31], since these SMs (ex. phenolics) may perform numerous protective functions and their amount is related to general or specific plant pathogenic resistance [2,28,32,33]. Some studies have reported that endophytic bacteria can affect plants’ secondary metabolism [30,34,35]. Moreover, recent studies have shown that plant-associated microorganisms can be antagonists to *H. fraxineus* and potentially serve as a tool for ash protection [21,36]. However, we have found no systematic studies related to the impact of different symbiotic bacteria on the formation of pathogenic resistance via SM production in *F. excelsior* as it relates to *H. fraxineus* infection. Additionally, studies show that different genotypes have different abilities to synthesize SM, therefore the genetic selection of resilient genotypes is appropriate [2,28,32,33].

Bacteria from *Paenibacillus* and *Pseudomonas* genera and their effect on trees have been studied in several works [37,38,39]. Bacteria from both genera have also been studied in relation to antimicrobial use [40,41,42], likely in part due to their potential for chitinase production [43,44]. Previously, we have shown that both bacteria may have a potential application in fighting *Phellinus tremulae*, rot-causing fungi pathogenic to aspen [37]. On the basis of these findings, we hypothesize that bacteria may boost SM production in *F. excelsior* seedlings, thus priming the tree’s metabolism and facilitating higher levels of resilience to *H. fraxineus*. Hence, this study aimed to assess the ability of *Paenibacillus* sp. and *Pseudomonas* sp. bacteria to regulate morphogenetic processes and activate protective SMs synthesis in three different European ash genotypes in vitro. This pilot study evaluates the possibilities of using plant-associated bacteria to improve the quality and resistance of forest replanting material.

## 2. Materials and Methods

### 2.1. Ash Genotypes

Seeds of three different genetic families of European ash were collected from a second-generation ash seed orchard in the Vytėnai regional division (Kaunas district, Lithuania) and Ubiškės regional division (Telšiai district, Lithuania). The origin sites of the genotypes are shown in Figure 1.

### 2.2. Microorganisms

Two different strains of bacteria were used—*Paenibacillus* sp. (closely related to *P. tundrae*, 99.06% Identity) and *Pseudomonas* sp. (closely related to *Ps. oryzihabitans*, 99.46% Identity). Both were isolated from aspen in vitro cultures. The isolation and identification of the bacteria were described previously [37]. Both bacterial strains were grown on a solid, low-salt, Lysogeny broth (LB) medium (Duchefa Biochemie, Haarlem, Netherlands) (pH ~ 7.0) at ~25 °C. In vitro cultures of the fungus *H. fraxineus* were established in LRCAF Forest Institute, using samples from *F. excelsior* trees in the form of lesions from naturally infected leaf petioles from a 40-year-old ash forest stand, damaged by ADB following the isolation procedure described by Kirisits et al. [45]. For *H. fraxineus* isolation, 10-mm-long petiole fragments containing pseudosclerotia were surface sterilized in 96% ethanol for 1 min, then in 4% NaOCl for 3 min, and then in 96% ethanol again for 30 s. Distilled water (dH_2_O) was used for rinsing in between the washing steps. Subsequently, the fragments were dried out for 2 min. Afterwards, pseudosclerotial layers were peeled off using sterile utensils. The remaining samples were placed onto a Petri dish containing sterile Hagem agar (malt extract 4 g/L, yeast extract 1 g/L, glucose 5 g/L, NH_4_Cl 0.5 g/L, KH_2_PO_4_ 0.5 g/L, MgSO_4_·7H_2_O 0.5 g/L, 1% FeCl_3_ 0.5 mL, 100 ppm thiamin 0.125 mL, dH_2_O to a liter; pH ~ 5.5) (components purchased from Duchefa Biochemie, Haarlem, The Netherlands). Isolates were grown at room temperature in the dark. All growing mycelium fragments were subsequently reisolated onto fresh Petri dishes with Hagem media. Suspected *H. fraxineus* isolates were screened and identified microscopically [46,47]. Additionally, further identification was caried out as described by Burokiene et al. [48] using *H. fraxineus* specific primers.

Fungus *H. fraxineus* was grown on a solid maltose medium containing maltose (20 g/L), yeast extract (8 g/L), tryptone (6 g/L), glucose (20 g/L) and gelrite (6 g/L) (pH 5.8) (Duchefa Biochemie) at ~25 °C (when not in use, all the microorganisms are kept as a glycerol stock at −20 °C).

### 2.3. Media

During the study seedlings were grown on Murashige and Skoog (MS) medium including vitamins, containing sucrose (20 g/L) and gelrite (4 g/L) (pH 5.8) (media and supplementary components purchased from Duchefa Biochemie). For the seed germination stage, 27 mL of medium was poured into each Petri dish (15 mm × 90 mm) and for the experiment, 30 mL of medium was poured into each clear plastic container (56 mm × 78 mm).

### 2.4. Seed Germination

Ash seeds were surface disinfected before planting. Firstly, seeds were placed in a liquid-permeable bag and immersed in 2.5% sodium hypochlorite solution for 10 min. After soaking, the bag was rinsed with warm water. The bag of seeds was then immersed in 75% ethanol for 3 min. Afterwards, the bag was transferred to a sterile Petri dish and filled with sterile distilled water, then subsequently soaked for 3 min (this wash was repeated 3 times in total). Later, the ash seed bag was transferred to a new sterile Petri dish and 0.2% silver nitrate solution was added until the seeds were all immersed. Seeds were soaked for 3 min. The bag with the seeds was then transferred to another Petri dish and soaked in sterile water for 3 min. The disinfected seeds were collected from the bag into a new sterile Petri dish. From here the seeds were taken one by one and, using sterile tweezers and scalpels, the germ was removed from each seed. The isolated embryo was then placed in a Petri dish with MS medium (5 germs per plate).

Germination took 2 weeks. After that, suitable (visually similar) seedlings were chosen for the experiment.

### 2.5. Inoculation with Bacteria and Fungi

For the first-stage inoculation, a day before the transfer of ash seedlings, the MS medium in the plastic containers was spot-inoculated with one of the bacteria using an inoculation needle with ~1.4 × 10^6^ cfu (colony forming units) of *Paenibacillus* sp. or ~6.4 × 10^6^ cfu of *Pseudomonas* sp. The bacteria used were taken from bacterial colonies formed on the surface of solid LB medium from overnight cultures.

The amount of cfu per bacterial scrape was determined in a separate experiment using the serial dilution technique.

The fungus *H. fraxineus* was used for the second-stage inoculation using a modified method [49]. A 7-mm-diameter sample from the edges of the mycelial growth was taken using a sterile cork borer, transferred to a container with the plant medium and placed in the middle, before the transfer of ash seedlings.

The survival of both bacteria and fungus was determined visually.

### 2.6. Explants in Bacteria/Fungus Inoculated Medium

Germinated plants from in vitro-germinated embryos (about 10 mm in length) were used as explants. They were planted in plastic containers (three explants in one container) with different experimental variants—control (uninoculated medium) and medium inoculated with either bacterial isolate. In the first culture stage, 84 explants for each variant were used in the experiment for family 64, and 40 explants for each variant for families 87 and 174 each. Plants were kept in a growth chamber for 4 weeks (25/20 °C under a 16/8 h photoperiod, white light, irradiance 30 μmol m^−2^ s^−2^).

After four weeks, the morphological parameters (shoot length, largest leaf width) of in vitro-grown plants were recorded and new leaves that had formed at the stage of cultivation with bacteria were harvested for biochemical studies. Then, explants were transferred to fungus-inoculated containers for the second-stage trials. Plants were grown for 4 additional weeks under the same conditions as described previously. Afterwards, plant vegetative growth parameters (rooting, shoot length and largest leaf width) were measured, and leaves were again collected for biochemical studies.

The general scheme of the conducted experiments is shown in Table 1.

### 2.7. Extract Preparation for Secondary Metabolite Analysis

First, 0.5 mg of fresh leaves were kept at −20 °C until analysis. Then, samples were homogenized using mortar and pestle. Then, the resulting material was shaken with 10 mL methanol (75%) for 24 h at 25 °C with a Kuhner Shaker X electronic shaker (Adolf Kühner AG, Birsfelden, Switzerland) at 150 rpm. The extracts were filtered using Whatman no. 1 filter paper, with a retention of 5–8 µm.

### 2.8. Quantification of Total Phenolic Compounds

Total phenolic content (TPC) was quantified using the *Folin–Ciocalteu* reagent per Singleton et al. [50]. Previously prepared extracts (0.1 mL) were mixed with 0.1 mL of the *Folin–Ciocalteu* reagent (2 N) and 2.5 mL of distilled water (dH_2_O). After 6 min, 0.5 mL of a 25% (*w*/*v*) NaCO_3_ solution was added to the mixture. This mixture was left at room temperature for 30 min. Then thee absorbance was measured using the Synergy HT Multi-Mode Microplate Reader (BioTek Instruments, Inc., Bad Friedrichshall, Germany) at 760 nm using 75% methanol as a blank. Phenol content was expressed as milligrams of chlorogenic acid per gram of fresh weight of leaves (mg CAE/g). The total phenolics were evaluated using a calibration curve y = 5.5358x − 0.0423 (R^2^ = 0.9975).

### 2.9. Quantification of Total Flavonoid Content

The total flavonoid content (TFC) in the extracts was quantified according to Striganavičiūtė et al. [37]. Briefly, the extract (1 mL) was mixed with 0.3 mL of 5% (*w*/*v*) NaNO_2_. After 5 min, 0.5 mL of 2% (*w*/*v*) AlCl_3_ was added. Then after an additional 6 min, the solution was neutralized with 0.5 mL of NaOH (1 M). Absorbance was recorded at 470 nm on the Synergy HT Multi-Mode Microplate Reader (BioTek Instruments, Inc.). TFC was expressed as milligrams of catechin per gram of fresh weight of leaves (mg CE/g). TFC was evaluated using a calibration curve y = 11.616x + 0.0634 (R^2^ = 0.9983).

### 2.10. Quantification of Photosynthesis Pigments

First, 0.2 g of fresh leaves was sampled for chlorophyll *a* and *b* (chl *a* and *b*) and carotenoid assays. For the determination of pigment content, the method described by D. Wettstein was used [51]. Fresh leaf matter was ground in acetone (VWR International, Radnor PA, USA) and filtered through Whatman no. 1 filter paper, with a retention of 5–8 µm. The content of carotenoids and chl *a* and *b* (mg/g) were estimated spectrophotometrically using the T80 UV-VIS spectrophotometer (PG Instruments, Leicestershire, UK) at wavelengths of 441, 662 and 644 nm (*D*), respectively. For the calculation of pigment content, the following models were used:(1)$$\mathrm{chl}\;a\;\left(\mathrm{mg}/g\right)=\frac{\left(9.784\times D_{662}-0.990\times D_{644}\right)\times V}{P\times 1000}$$
(2)$$\mathrm{Carotenoids}\;\left(\mathrm{mg}/g\right)=\frac{\left(4.695\times D_{441}-0268\times \left(\mathrm{chl}\;a+\mathrm{chl}\;b\right)\right)\times V}{P\times 1000}$$
(3)$$\mathrm{chl}\;b\;\left(\mathrm{mg}/g\right)=\frac{\left(21.426\times D_{644}-4.650\times D_{662}\right)\times V}{P\times 1000}$$
where *V* = extract volume (mL) and *P* = fresh leaf matter (g).

### 2.11. Statistical Data Analysis

Each measurement for SMs and pigments was calculated as an average of 3 technical replicates and 3 biological relicates. A two-tailed Welch’s *t*-test, intended to compare samples with possibly unequal variances, was conducted to calculate the probability that the means of the different variants are equal [52] (Microsoft Excel). Experimental variants not infected with the fungus were compared with the control without the fungus, and variants using the fungus were compared with the control variant also infected with the fungus.

## 3. Results

### 3.1. Isolation and Identification of H. fraxineus

The isolated fungus from naturally infected *F. excelsior* trees was identified as *H. fraxineus*, both by morphological and DNA results (Figure 2).

### 3.2. Effect of the Studied Microorganisms on Rooting of Different Ash Genotypes

The data analysis showed that different ash genotypes showed different responses to inoculation with studied microorganisms, regarding the percentage of rooted explants in the three investigated *F. excelsior* genotypes (Table 2).

The data analysis showed that all tested *F. excelsior* genotypes did not have a positive response to the first-stage and second-stage treatment with either bacterium *Pseudomonas* sp. or bacterium *Paenibacillus* sp., in regard to rooting (Table 2). However, genotype 87 differed from two others in its response to the first-stage bacterial inoculation. This genotype showed a negative response to both bacteria. The combination of *Pseudomonas* sp. and *H. fraxineus* had a negative effect on rooting for genotype 174.

### 3.3. Effect of Bacterial Inoculation on Ash Shoot Growth and Leaf Biochemistry

The direct effect of nutrient medium inoculation with *Paenibacillus* sp. or *Pseudomonas* sp. bacteria on shoot development of different *F. excelsior* genotypes is shown in Figure 3 (stage 1). In European ash genotype 64, neither average shoot length (Figure 3a) nor leaf width (Figure 3b) was affected by the *Paenibacillus* sp. bacteria, while *Pseudomonas* sp. had a negative effect on shoot length. A different situation was observed for genotype 87, where the shoot and leaf development parameters were significantly decreased by both bacteria. In genotype 174, shoot length was negatively affected by *Pseudomonas* sp. The statistically significant relatively strongest negative impact on shoot length was observed for *Pseudomonas* sp. bacteria for all tested European ash genotypes (Figure 3a).

The bacterial effects on the chlorophyll pigment contents in *F. excelsior* leaves are presented in Figure 4. The obtained results show that treatment with both *Paenibacillus* sp. and *Pseudomonas* sp. bacterium significantly increased the chlorophyll *a/b* ratio in genotype 64 (Figure 4a). Meanwhile, the chlorophyll *a/b* ratio was significantly decreased in response to both bacteria in genotype 87 (Figure 4a), while genotype 174 was not affected.

Similar effects of bacterium *Pseudomonas* sp., as induced on the chlorophyll *a/b* ratio, was also found in the carotenoid content (Figure 4b) in European ash genotypes 64 and 87.

However, while *Paenibacillus* sp. had a negative effect on the carotenoid content in genotype 87, *Pseudomonas* sp. facilitated a positive increase.

Additionally, with respect to the carotenoid content, the response of European ash genotype 174 to *Pseudomonas* sp. was different from 64 and 87 genotypes—carotenoid content decreased with *Pseudomonas* sp. bacterium (Figure 4b).

The TPC (Figure 5a) was highly increased in response to the bacterium *Pseudomonas* sp. in European ash genotype 87. Meanwhile, the same bacterium had a slight positive effect on total phenol content in genotype 64. Moreover, treatment with *Paenibacillus* sp. significantly increased the total phenol content in European ash genotypes 64 and 87.

The results on TFC presented in Figure 5b reveal that European ash genotype 87 showed a significantly positive response to bacterium *Paenibacillus* sp., and genotype 174 showed a positive response to bacterium *Pseudomonas* sp. Moreover, genotype 64 showed a positive response to both *Paenibacillus* sp. and *Pseudomonas* sp. bacteria.

### 3.4. Long-Term Effect of the Studied Bacteria on Ash Shoot Growth and Leaf Biochemistry after Inoculation with H. fraxineus

The data shown in Figure 6 represents the long-term effect of *Paenibacillus* sp. or *Pseudomonas* sp. bacteria on shoot development of different *F. excelsior* genotypes. The results of second-stage trials were obtained after the explants from the bacterium-inoculated nutrient medium were transferred and cultured on either sterile or fungus-inoculated mediums (stage 2).

It was found that there were no statistically significant differences in average shoot length (Figure 6a), nor leaf width (Figure 6b) in the effect of the bacterium *Paenibacillus* sp., neither in the sterile second-stage medium nor on the nutrient medium inoculated with *H. fraxineus.* However, the average shoot parameters in European ash genotype 87 decreased after being inoculated with the bacterium *Pseudomonas* sp. on the nutrient medium inoculated with *H. fraxineus* (Figure 6a,b).

Moreover, the *Pseudomonas* sp. bacterial effect on the leaf development of genotype 87 caused a significant leaf width decrease in both variants (Figure 6b).

### 3.5. Long-Term Bacterial Inoculation Effect on Ash Leaf Biochemistry after Inoculation with H. fraxineus

The long-term bacterial effects on the chlorophyll and carotenoid content in *F. excelsior* leaves are shown in Figure 7.

No consistent effect on photosynthetic pigments was observed in either variant. Long-term effects on the chlorophyll *a/b* ratio of both *Paenibacillus* sp. and *Pseudomonas* sp. inoculation were significantly lower in the fungus-free medium of European ash genotype 87. It was observed that the response of genotype 64 to *Paenibacillus* sp. bacteria was negative, and the response of genotype 174 to the same bacteria was positive (Figure 7a). Moreover, the *Pseudomonas* sp. bacterial effect on the chlorophyll a/b ratio of genotype 64 caused a significant decrease in the nutrient medium inoculated with *H. fraxineus* (Figure 7a). Additionally, genotype 174 was negatively affected by *H. fraxineus* if the explants were inoculated with both bacterium *Paenibacillus* sp. and *Pseudomonas* sp.

Long-term bacterial effects on carotenoid content were best seen if the medium of the second stage was not inoculated with *H. fraxineus* (Figure 7b). It was noted that the long-term effect of *Paenibacillus* sp. increased carotenoid content in genotypes 64 and 87. Meanwhile, *Pseudomonas* sp. increased carotenoid content in genotype 64. Only *Paenibacillus* sp. had a significant effect on carotenoid content in *H. fraxineus* inoculated shoots; it was positive in genotype 174.

The long-term bacterial effects on SM content (phenols and flavonoids) in *F. excelsior* leaves are shown in Figure 8.

*Paenibacillus* sp. and *Pseudomonas* sp. led to increased concentrations of stress-resilience indicators, including not only previously described carotenoids (Figure 8b) but also phenolic and flavonoid compounds in fungus-free media (Figure 8). In fungus-free genotype 87, TPC increased under the long-term effect of *Pseudomonas* sp., and decreased in genotype 174 after *Paenibacillus* sp. treatment (Figure 8a). Both bacteria had a positive effect on TPC of genotype 64 in this variant.

When inoculated with *H. fraxineus*, *Pseudomonas* sp. inoculated shoots had higher TPC than their respective controls in European ash genotypes 64 and 87, while in genotype 174 previously inoculated with *Paenibacillus* sp., TPC significantly decreased. Meanwhile, in seedlings of genotype 87 on the nutrient medium inoculated with *H. fraxineus*, after the treatment with the *Paenibacillus* sp. bacterium, TPC increased (Figure 8a).

Both bacteria increased TFC in genotype 64, while *Pseudomonas* sp. increased it in genotype 87 in fungus-free media; however, after treatment with the *Paenibacillus* sp. bacterium, TFC decreased in genotypes 87 and 174 in fungus-free media (Figure 8a). A statistically significant increase in TFC was determined in *H. fraxineus*-inoculated genotypes 64 and 87 after treatment with the *Pseudomonas* sp. bacterium. Meanwhile, *Paenibacillus* sp. had a significant positive impact on TFC of genotype 87 on the nutrient medium inoculated with *H. fraxineus*.

## 4. Discussion

*H. fraxineus* causes major damage in European forests, thus the biocontrol of this pathogen is of great importance [53,54]. It has been shown that some European ash genotypes may be somewhat genetically resilient [11,23,24,25,26,27,55,56]. However, we propose that a plant’s secondary metabolism can play a part in ash resistance to *H. fraxineus* as well. Specifically, the amount of phenolic the trees produce can be used as an indicator [2,57]. Since the production of phenolics is also heritable, genetic selection for the most suitable material for ash reforestation is fitting.

SMs such as phenols and flavonoids are often associated with a plant’s response to both biotic and abiotic stress [28,30,31,32,58,59,60,61]. Higher production of compounds in both of these groups is indicative of systemic resistance due to increases in the chemical defensive response [32,37,62].

Inoculation with microorganisms can influence a plant’s response to varied negative factors [32,37,63,64,65,66,67,68]. Furthermore, microorganisms have an intimate relationship with a plant’s metabolism, possibly serving as regulators of interactions [30]. Thus, in addition to the genetic selection of *F. excelsior* resilient to *H. fraxineus* infection, plant-associated bacteria may also be used to boost ash resilience by priming the secondary metabolism apparatus, eliciting induced systemic resistance.

Recent studies have reported ash endophytes that may contribute to tree vitality through antagonisms against *H. fraxineus* in vitro [21,36,53,69,70].

Moreover, research suggests that bacteria can facilitate enhanced SM production [37,65,71,72,73,74]. Microorganisms affect their host metabolism and protein synthesis in a multitude of ways [75,76], thus pinpointing the exact nature of the bacteria–pathogen–tree interaction is difficult. However, it has been proposed that endophytes may affect a plant’s secondary metabolism in one of three ways: An increase in plant material, which produces the metabolite, the alteration of metabolic pathways or optimization of the production process [77].

During our study, no significant increase in plant morphometric parameters was observed. However, in ash genotype 64, both bacteria elicited an increase in the echlorophyll a/b ratio, carotenoid content, TPC and TFC in the first-stage trial, and carotenoid content, TFC and TPC in the second-stage trials. Moreover, *Pseudomonas* sp. inoculated seedlings exhibited increased phenolics after *H. fraxineus* infection too. Increases in chlorophyll content are often linked to enhanced plant health [68,78], while carotenoid content is usually linked to stress, hormone production and antioxidative activity [79,80,81,82,83]. This could potentially indicate that the tested bacteria induce low amounts of stress, thus in return, their host plant upregulates SM production.

Tested bacteria and other ash genotypes did not appear to interact in the same manner as genotype 64. However, in the trial of ash genotype 87, *Pseudomonas* sp. elicited an increase in TPC and TFC, with or without the additional *H. fraxineus* infection. Interestingly, it decreased rooting when seedlings were not inoculated with the fungus. Additionally, *Paenibacillus* sp. also similarly affected family 87, however, only post pathogen infection.

It has been proposed that *H. fraxineus* attacks European ash at least in part via the phytotoxic metabolite viridiol [55,84]. Cleary et al. found that less-susceptible ash trees had higher amounts of secoiridoids [55]. On the other hand, Nemesio-Gorriz et al. reported it might be coumarins that are responsible for European ash resilience to *H. fraxineus* [85]. Their research explores the chemical defense of tested trees. They have found that susceptible trees and resilient trees differed in their SM profile. Interestingly, the bark extracts from resilient trees affected the pathogen negatively as did the two isolated coumarins fraxetin and esculetin, which were linked to reduced susceptibility to *H. fraxineus*. It is possible plant-associated bacteria may upregulate the production of these metabolites (references therein [86]).

Previously, we tested these same bacterial isolates in an analogue experiment on aspen microshoots in vitro using the pathogen *Phellinus tremulae—*aspen root rot causing fungus [37]. The results were also genotype-dependent, as the effects of both pathogen and tested bacteria varied within a genetic family. It was observed that *Pseudomonas* sp. affected the morphometric parameters and pigment content of some aspen positively, while *Paenibacillus* sp. demonstrated a large increase in TPC and TFC in pathogen-infected shoots.

It is also noteworthy that during this study, both *Pseudomonas* sp. and *Paenibacillus* sp. affected ash growth negatively, as seen in the rooting data, shoot and leaf growth. This too was noted in our previous study, whereby both bacteria had a negative effect on hybrid aspen and not *Populus tremula*. In addition, *Pseudomonas* sp. had a synergistic effect with *H. fraxineus—*seedlings inoculated with both bacterium and fungus facilitated significantly lower rooting.

## 5. Conclusions

Bacterial (*Pseudomonas* sp. and *Paenibacillus* sp.) interaction with ash (*Fraxinus excelsior*) seedlings and the pathogenic fungus *Hymenoscyphus fraxineus* was investigated. Plant-associated bacteria were shown to facilitate increased phenolics and photosynthesis pigment content (*Pseudomonas* sp. was especially effective), and hence help prime the plants for subsequent pathogenic infection. It is noteworthy that the effect on ash seedlings was dependent on tree genotype, thus demonstrating, in addition to using genetic selection of resilient ash replanting material, utilizing bacteria to boost the production of phenolics is also appropriate.

Future studies will test the effect of ash inoculation under ex vitro conditions, using different inoculation techniques and bacterial concentrations to develop a methodology applicable under field conditions, as well as the identification of specific secondary metabolites.

## Figures and Tables

**Figure 1 microorganisms-09-02504-f001:**
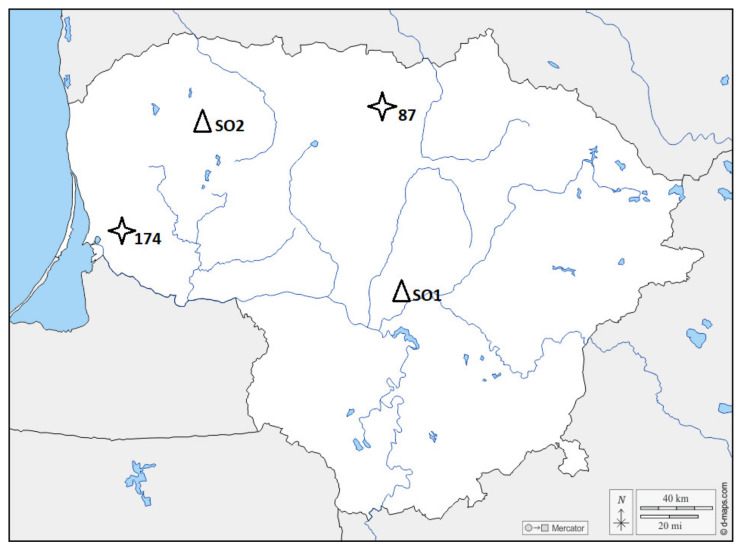
Map of Lithuania showing the seed orchard (marked with ∆SO1 for 174 and 87 genetic families and ∆SO2 for the 64 genetic family on the map) and origin sites of the genotypes (marked with an asterisk on the map) used for the study. The origin site of the 64 genetic family cannot be identified.

**Figure 2 microorganisms-09-02504-f002:**
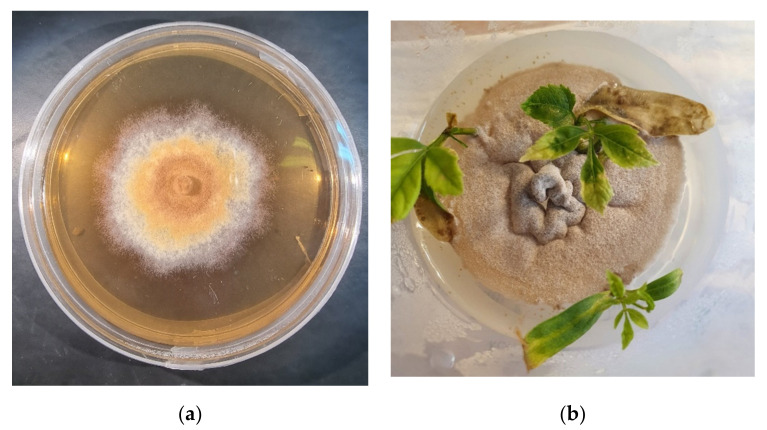
Fungus *H**ymenoscyphus*
*fraxineus*: (**a**) In a pure culture on maltose nutrient medium; (**b**) with ash microshoots post inoculation.

**Figure 3 microorganisms-09-02504-f003:**
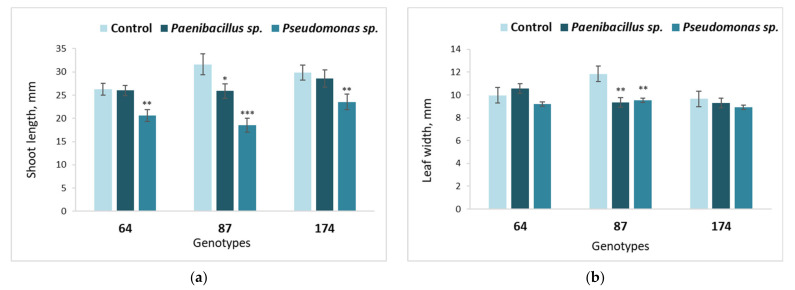
Average shoot length (**a**) and leaf width (**b**) of different ash genotypes (64, 87 and 174), inoculated with *Paenibacillus* sp. or *Pseudomonas* sp. bacteria. Statistically significant differences from the bacterium-free control variant are indicated (for each genotype separately): * *p* < 0.05; ** *p* < 0.01; *** *p* < 0.001.

**Figure 4 microorganisms-09-02504-f004:**
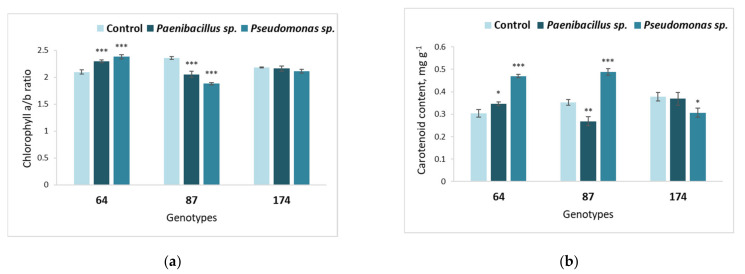
Average chlorophyll a/b ratio (**a**) and carotenoid content (**b**) in the leaves of different *F. excelsior* genotypes (64, 87 and 174), inoculated with *Paenibacillus* sp. and *Pseudomonas* sp. bacteria. Statistically significant differences from the bacterium-free control variant are indicated (for each genotype separately): * *p* < 0.05; ** *p* < 0.01; *** *p* < 0.001.

**Figure 5 microorganisms-09-02504-f005:**
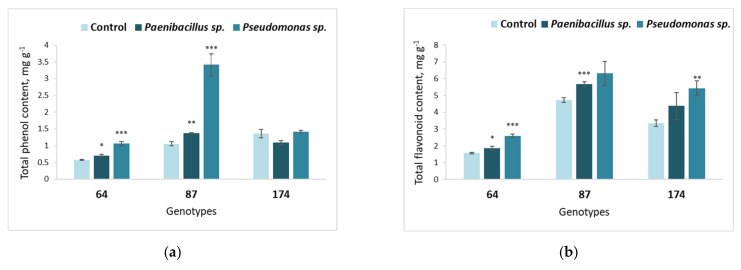
Average TPC (**a**) and TFC (**b**) in the leaves of different *F. excelsior* genotypes (64, 87 and 174), inoculated with *Paenibacillus* sp. and *Pseudomonas* sp. bacteria. Statistically significant differences from the bacterium-free control variant are indicated (for each genotype separately): * *p* < 0.05; ** *p* < 0.01; *** *p* < 0.001.

**Figure 6 microorganisms-09-02504-f006:**
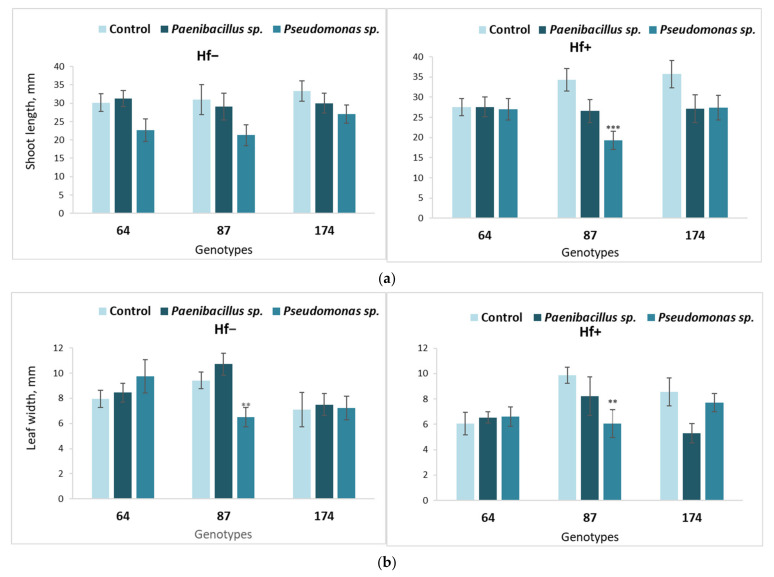
Average shoot length (**a**) and leaf width (**b**) of different *F. excelsior* genotypes (64, 87 and 174), of which explants were first cultured on the nutrient medium inoculated with *Paenibacillus* sp. or *Pseudomonas* sp. bacteria, then transferred onto fresh nutrient medium, either sterile (‘Hf−’) or inoculated with the fungus *H. fraxineus* (‘Hf+’). Statistically significant differences from the control variant, which was not treated with bacteria during the first culture stage, are indicated (for each genotype separately): ** *p* < 0.01; *** *p* < 0.001.

**Figure 7 microorganisms-09-02504-f007:**
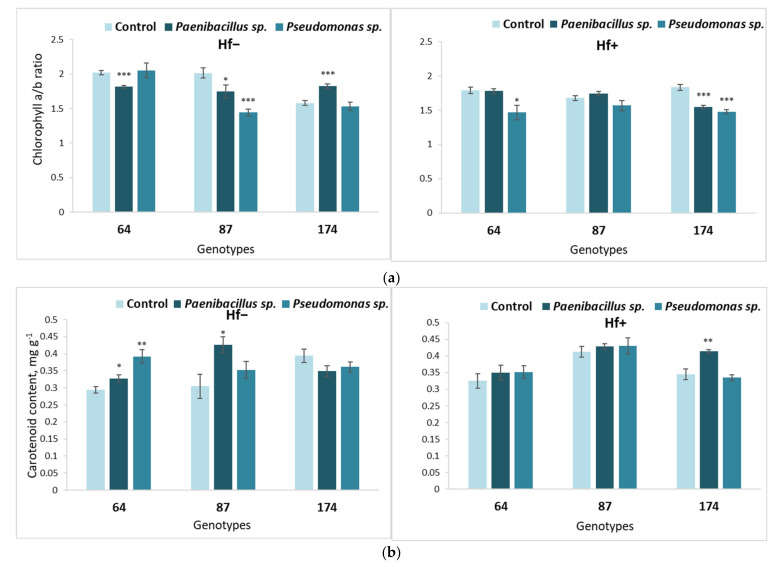
Average chlorophyll *a/b* ratio (**a**) and carotenoid content (**b**) in the leaves of different *F. excelsior* genotypes (64, 87 and 174), of which explants were first cultured on the nutrient medium inoculated with *Paenibacillus* sp. or *Pseudomonas* sp. bacteria, then transferred onto fresh nutrient medium, either sterile (‘Hf−’) or inoculated with the fungus *H. fraxineus* (‘Hf+’). Statistically significant differences from the control variant, which was not treated with bacteria during the first culture stage, are indicated (for each genotype separately): * *p* < 0.05, ** *p* < 0.01, *** *p* < 0.001.

**Figure 8 microorganisms-09-02504-f008:**
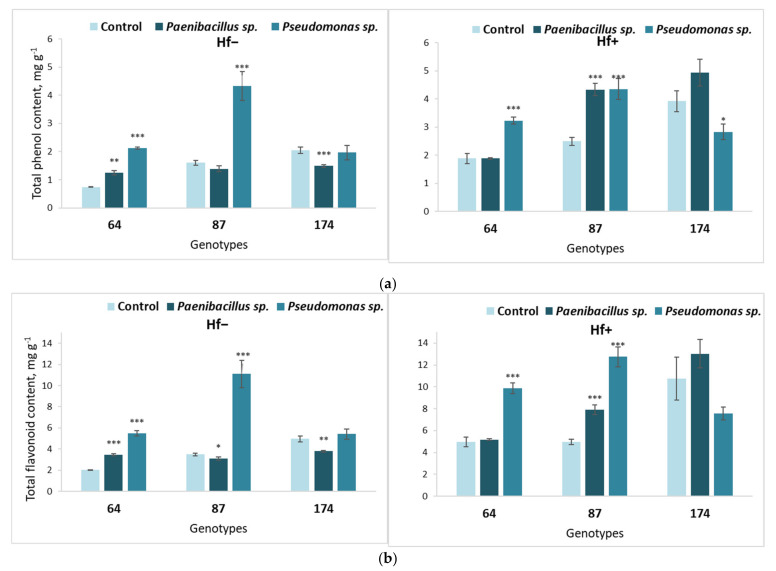
Average TPC (**a**) and TFC (**b**) in the leaves of different *F. excelsior* genotypes (64, 87 and 174), of which explants were first cultured on the nutrient medium inoculated with *Paenibacillus* sp. or *Pseudomonas* sp. bacteria, then transferred onto fresh nutrient medium, either sterile (‘Hf−’) or inoculated with the fungus *H. fraxineus* (‘Hf+’). Statistically significant differences from the control variant, which was not treated with bacteria during the first culture stage, are indicated (for each genotype separately): * *p* < 0.05; ** *p* < 0.01; *** *p* < 0.001.

**Table 1 microorganisms-09-02504-t001:** Scheme of the inoculation experiments with different microorganisms (bacteria *Paenibacillus* sp. and *Pseudomonas* sp. and fungus, *Hymenoscyphus fraxineus*) during two culturing stages.

Experimental Variant	First Culture Stage (Four Weeks)	Second Culture Stage ((Four Weeks)
Control	-	-
+Hf.	-	*H.* *fraxineus*
Paen	*Paenibacillus* sp.	-
Paen + Hf	*Paenibacillus* sp.	*H.* *fraxineus*
Pseu	*Pseudomonas* sp.	-
Pseu + Hf	*Pseudomonas* sp.	*H.* *fraxineus*

**Table 2 microorganisms-09-02504-t002:** Percentage of rooted explants in the in vitro cultures of three *Fraxinus excelsior* genotypes after second stage of culturing with different microorganisms. In the first stage, the nutrient medium was inoculated with either *Paenibacillus* sp. (‘Paen’) or *Pseudomonas* sp. (‘Pseu’) bacteria. For the second stage, explants from each variant were transferred onto fresh medium, either sterile or inoculated with fungus *H*. *fraxineus* (‘+Hf’). The control group of explants was cultured on sterile media during both stages.

Experimental Variant	European Ash (*Fraxinus excelsior* L.)		
	64	87	174
Control	56.52 ± 10.57	91.67 ± 8.33	100 ± 0
+Hf	58.82 ± 12.30	66.67 ± 12.60	100 ± 0
Paen	77.78 ± 10.08	50 ± 13.87 *	70 ± 15.28
Paen + Hf	62.50 ± 10.09	77.78 ± 14.99	72.73 ± 14.08
Pseu	80.95 ± 8.78	57.14 ± 13.73 *	66.67 ± 16.67
Pseu + Hf	91.30 ± 6.01	61.11 ± 11.82	60 ± 16.33 *

Statistically significant differences from the control variant (for each genotype separately) are indicated: * *p* < 0.05.

## Data Availability

Complete study data are available upon request.

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
