# Peer review of "Priming of Resistance-Related Phenolics: A Study of Plant-Associated Bacteria and Hymenoscyphus fraxineus"

_microorganisms, 2021, doi:10.3390/microorganisms9122504_

Round 1

Reviewer 1 Report

The authors studied the effect of two bacterial strains on priming of antimicrobial phenolics in ash seedlings. Especially against the background of the severe ash dieback in Europe, this is an interesting paper dealing with the induction of a system resistance in ash by inoculation with bacterial strains. The paper is well written, the experimental design is appropriate and the results are clear and unambiguous. There are only some minor concerns to be revised.

The abstract is well structured by it should also be mentioned that the results were achieved under in vitro conditions. In this context, I suggest to add a paragraph in the discussion to reflect the transferability of the results to older ash plants under greenhouse conditions or grown in the field.

It is important to indicate the name of the used bacterial strains (from the given reference). The described effects are known to be strain-specific. The strain names should be used throughout the manuscript.

Please also name the H. fraxineus isolate. Further, the identity of the fungus should -if possible- additionally be proven by a qPCR or even better by sequencing of the ITS rRNA region. It requires just a PCR and the sequencing of the amplicon by a company.

 “Both bacterial strains” instead of “Both species of bacteria”

I am confused about the cultivation time. Was it 4 weeks as given in the text or 6 weeks as indicated in Table 1?

Welch’s t-test: Were the sample populations tested for normal distribution? Why did you use a pairwise comparison for the analysis of six variants? I would expect an ANOVA with a suitable post-hoc test or a comparable non-parametric test. This would also demonstrate differences between the variants and not only against the control.

Paragraph 3.3. Please specify if leaves from stage 1 or 2 were used for the analysis. I guess it was stage 1.

Author Response

Dear reviewer

We are very grateful for the valuable reviewers’ comments on our manuscript “Priming of Antimicrobial Phenolics: A Study of Plant-Associated Bacteria and Hymenoscyphus fraxineus”. We have changed the title of the manuscript according to the Academic Editor\s suggestion to “Priming of Resistance-Related Phenolics: A Study of Plant-Associated Bacteria and Hymenoscyphus fraxineus”. Also, we have changed the manuscript according to remarks of the reviewers and we have uploaded the revised manuscript. Information on all performed modifications is provided in detail with answers to the remarks of the reviewers. We very much hope that the performed changes improved the manuscript, and the revised version will be accepted for publishing in the Special Issue “Plant Associated Bacteria, so Different and so Similar: From Pathogens to Symbionts and to Biological Control Agents” of the journal Microorganisms.

We sincerely thank the Reviewers for the review of our manuscript. Please find the comments below and changes in the manuscript.

Answers to reviewer 1:

  • The abstract is well structured by it should also be mentioned that the results were achieved under in vitro conditions.

Reply: According to the reviewer’s suggestion additional information was added (lines 16, 80).

  • In this context, I suggest to add a paragraph in the discussion to reflect the transferability of the results to older ash plants under greenhouse conditions or grown in the field.

Reply: The suggested information was added to the Conclusions section (line 435-438).

  • It is important to indicate the name of the used bacterial strains (from the given reference). The described effects are known to be strain-specific. The strain names should be used throughout the manuscript.

Reply: The reviewers’ suggestion would be appropriate if the strains would be conclusively identified. However, 16S rRNA gene fragment sequencing results don’t always yield species level identification, specifically in highly variable genera, like Pseudomonas. Based on current guidelines for identifying bacteria at species level (100% query coverage and the next bacteria in line suggested as a homologue is at least ≤0.5% Identity points lower), neither one of the strains used in this work can be thus identified. Hence, in this instance, so as not to claim definitive knowledge on the strains we worked with, we chose to omit species identifiers, except for lines 99-101.

  • Please also name the fraxineus isolate.

Reply: Again, due to the identification method used (see lines 119-121), we cannot claim a conclusive knowledge on the strain of the H. fraxineus isolate used in this study, thus strain specific identification would be negligible.

  • Further, the identity of the fungus should -if possible- additionally be proven by a qPCR or even better by sequencing of the ITS rRNA region. It requires just a PCR and the sequencing of the amplicon by a company.

Reply: We amended the Materials and Methods subsection 2.2. (lines 119-121, as well as 229-231).

  • “Both bacterial strains” instead of “Both species of bacteria”.

Reply: Line 102 was amended, as suggested.

  • I am confused about the cultivation time. Was it 4 weeks as given in the text or 6 weeks as indicated in Table 1?

Reply: We are grateful to the reviewer for noticing our oversight. The information on cultivation times was amended where relevant (lines 172, Table 1).  

  • Welch’s t-test: Were the sample populations tested for normal distribution? Why did you use a pairwise comparison for the analysis of six variants? I would expect an ANOVA with a suitable post-hoc test or a comparable non-parametric test. This would also demonstrate differences between the variants and not only against the control.

Reply: The sample populations were checked for normal distribution. Pairwise comparison was used, because we were mainly interested in comparing differences between the Control group and the relevant Inoculated group, the differences between two or more inoculated groups were of little interest to us at this time. Welch’s t-test was selected because the populations were of varied sizes, and relevant recent literature sources were suggestive of the adequateness of this method (R.M. West, 2021, Best practice in statistics: Use the Welch t-test when testing the difference between two groups; doi: 10.1177%2F0004563221992088). However, we will take the reviewer’s suggestion under advisement in future works.

  • Paragraph 3.3. Please specify if leaves from stage 1 or 2 were used for the analysis. I guess it was stage 1.

Reply: We made corrections to the article based on this suggestion, to make it clearer and more reader-friendly (lines 253, 296).

Sincerely,

Vaida Sirgedaitė-Šėžienė, Ph.D

Corresponding author: e-mail: [email protected]

Laboratory of Forest Plant Biotechnology

Institute of Forestry, Lithuanian Research Centre for Agriculture and Forestry

Reviewer 2 Report

Brief Summary

The manuscript microorganisms-1472606 report interesting results on the use of Paenibacillus sp. and Pseudomonas sp. in Fraxinus excelsior to boost the production of secondary metabolites active against Hymenoscyphus fraxineus. The authors tested three different F. excelsior genotypes and evaluated inoculation effects under gnotobiotic conditions. The secondary metabolites accumulation was estimated by total phenol (TPC), total flavonoid (TFC), carotenoid contents and chlorophyll a/b ratio. Morphometric growth parameters were also estimated.

Broad comments

The manuscript is well prepared and the findings interesting. I would change some aspects to improve the quality.

  • Introduction: The Introduction correctly places the study in the context with an explicit statement of the purpose of the study. However, the specific hypotheses being tested should be added.
  • Materials and Methods: The authors described with sufficient detail the methods used. Some details on microbial inoculation should be provided, see specific comments on pdf file attached.
  • Results: The results description is clear. I have minor suggestions, see specific comments on pdf file attached.
  • Discussion: Authors correctly discussed the results from the perspective of previous studies in the broadest context possible. I would provide more details on study limitations and future studies that should be carried out. See specific comments on pdf file attached.
  • Conclusions: The section is appropriate and in line with the findings got.

Specific comments

Please, see the pdf file attached.

Author Response

Dear reviewer

We are very grateful for the valuable reviewers’ comments on our manuscript Priming of Antimicrobial Phenolics: A Study of Plant-Associated Bacteria and Hymenoscyphus fraxineus. We have changed the title of the manuscript according to the Academic Editor\s suggestion to “Priming of Resistance-Related Phenolics: A Study of Plant-Associated Bacteria and Hymenoscyphus fraxineus”. Also, we have changed the manuscript according to remarks of the reviewers and we have uploaded the revised manuscript. Information on all performed modifications is provided in detail with answers to the remarks of the reviewers. We very much hope that the performed changes improved the manuscript, and the revised version will be accepted for publishing in the Special Issue “Plant Associated Bacteria, so Different and so Similar: From Pathogens to Symbionts and to Biological Control Agents” of the journal Microorganisms.

We sincerely thank the Reviewers for the review of our manuscript. Please find the comments below and changes in the manuscript.

Answers to reviewer  2:

1) The Introduction correctly places the study in the context with an explicit statement of the purpose of the study. However, the specific hypotheses being tested should be added.

Reply: The required corrections were added (lines 13, 21-22, 76-82).

2) The authors described with sufficient detail the methods used. Some details on microbial inoculation should be provided, see specific comments on pdf file attached.

Reply: We amended subsection 2.5 for clarity as much as was possible at this time. As mentioned, the serial dilution was done as a separate experiment. The inoculation was done with a bacterial scrape, thus just the approximate amount of cfu’s is indicated. Furthermore, for this study we only considered the area of mycelial growth, not the quantity. The survival of the fungus can be seen in Figure 2b (information was added in line 164). We will take the reviewer’s further suggestions under advisement for future studies.

 Also, amendments were made to lines 102-104, 122-125, 159, 183, 220-221, as per the suggestions in the pdf file.

We are very grateful to the reviewer for drawing our attention to the formulas. We’ve previously made an editing error and have now corrected it. This correction doesn’t affect the results, or the conclusions drawn from them, as the mistake was merely a typographical one.

3) The results description is clear. I have minor suggestions, see specific comments on pdf file attached.

Reply: Reviewer’s suggestions were taken under advisement and corrections were made where necessary (lines 232, 237, 244, Figure 2, 3-8).

4) Authors correctly discussed the results from the perspective of previous studies in the broadest context possible. I would provide more details on study limitations and future studies that should be carried out. See specific comments on pdf file attached.

Reply: According to reviewers’ suggestion, changes have been made to the manuscript (lines 361, 371, 373, 377, 435-438).

Moreover, in depth information about induced systemic resistance was not added at this time. It’s both due to the fact, that our current results do not indicate which type of resistance was facilitated, as well as the fact that systemic acquired resistance is a form of induced systemic resistance (Kamle. M. et al., 2020, Systemic Acquired Resistance (SAR) and Induced Systemic Resistance (ISR): Role and Mechanism of Action Against Phytopathogens in Fungal Biotechnology and Bioengineering, p. 457-470; doi: 10.1007/978-3-030-41870-0_20).

Sincerely,

Vaida Sirgedaitė-Šėžienė, Ph.D.

Corresponding author: e-mail: [email protected]

Laboratory of Forest Plant Biotechnology

Institute of Forestry, Lithuanian Research Centre for Agriculture and Forestry